# Simulating rigid head motion artifacts on brain magnitude MRI data–Outcome on image quality and segmentation of the cerebral cortex

Hampus Olsson[1]*, Jason Michael Millward[1,2], Ludger Starke[1], Thomas Gladytz[1], Tobias Klein[1], Jana Fehr[3], Wei-Chang Lai[3], Christoph Lippert[3,4], Thoralf Niendorf[1,2], Sonia Waiczies[1,2]*

1 Max-Delbrück-Center for Molecular Medicine in the Helmholtz Association (MDC), Berlin Ultrahigh Field Facility (B.U.F.F.), Berlin, Germany, 2 Experimental and Clinical Research Center, A Joint Cooperation Between the Charité Medical Faculty and the Max-Delbrück-Center for Molecular Medicine in the Helmholtz Association (MDC), Berlin, Germany, 3 Digital Health & Machine Learning Group, Hasso Plattner Institute for Digital Engineering, Potsdam, Germany, 4 Hasso Plattner Institute for Digital Health at Mount Sinai, Icahn School of Medicine at Mount Sinai, New York, NY, United States of America

* n.hampus.olsson@gmail.com (HO); sonia.waiczies@mdc-berlin.de (SW)

**Data Availability Statement:** MRI data with simulated (original and modified) motion artifacts are available here: https://openneuro.org/datasets/

## Abstract

Magnetic Resonance Imaging (MRI) datasets from epidemiological studies often show a lower prevalence of motion artifacts than what is encountered in clinical practice. These artifacts can be unevenly distributed between subject groups and studies which introduces a bias that needs addressing when augmenting data for machine learning purposes. Since unreconstructed multi-channel k-space data is typically not available for population-based MRI datasets, motion simulations must be performed using signal magnitude data. There is thus a need to systematically evaluate how realistic such magnitude-based simulations are. We performed magnitude-based motion simulations on a dataset (MR-ART) from 148 subjects in which real motion-corrupted reference data was also available. The similarity of real and simulated motion was assessed by using image quality metrics (IQMs) including Coefficient of Joint Variation (CJV), Signal-to-Noise-Ratio (SNR), and Contrast-to-Noise-Ratio (CNR). An additional comparison was made by investigating the decrease in the Dice-Sørensen Coefficient (DSC) of automated segmentations with increasing motion severity. Segmentation of the cerebral cortex was performed with 6 freely available tools: FreeSurfer, BrainSuite, ANTs, SAMSEG, FastSurfer, and SynthSeg+. To better mimic the real subject motion, the original motion simulation within an existing data augmentation framework (TorchIO), was modified. This allowed a non-random motion paradigm and phase encoding direction. The mean difference in CJV/SNR/CNR between the real motion-corrupted images and our modified simulations (0.004±0.054/-0.7±1.8/-0.09±0.55) was lower than that of the original simulations (0.015±0.061/0.2±2.0/-0.29±0.62). Further, the mean difference in the DSC between the real motion-corrupted images was lower for our modified simulations (0.03±0.06) compared to the original simulations (-0.15±0.09). SynthSeg+ showed the highest robustness towards all forms of motion, real and simulated. In conclusion, reasonably realistic synthetic motion

ds004795/versions/1.0.0. Python code for the modified motion simulations is available here: https://github.com/OlssonHampus02/motion-simulations. TorchIO, used to perform the original simulations is available here: https://github.com/fepegar/torchio. The original MR-ART dataset is available here https://openneuro.org/datasets/ds004173/versions/1.0.2.

**Funding:** This project was funded by the German Federal Ministry of Education and Research (BMBF, URL: https://www.bundesregierung.de/breg-en/federal-government/ministries/federal-ministry-of-education) within the project 'Syreal' (Grant No.01/S21069A). The funding was awarded to SW and CL. The funders had no role in study design, data collection and analysis, decision to publish, or preparation of the manuscript.

**Competing interests:** I have read the journal's policy and the authors of this manuscript have the following competing interests: HO is currently employed by Philips Healthcare. Remaining authors have declared that no competing interests exist. This does not alter our adherence to PLOS ONE policies on sharing data and materials.

artifacts can be induced on a large-scale when only magnitude MR images are available to obtain unbiased data sets for the training of machine learning based models.

## Introduction

Image artifacts due to rigid head motion are a very common cause of poor diagnostic image quality in neurological MRI [1]. Motion artifacts typically manifest as ghosting, blurring, and ringing in the image [2]. Several prospective and retrospective motion correction techniques exist, but these are seldom employed in routine neurological practice [3].

Motion simulations are an important form of data augmentation, to enhance the variability of training data for machine learning applications and improve the robustness of trained algorithms to deal with clinically realistic data [4]. Increasingly, researchers are performing motion simulations in a 'pseudo' k-space by applying a Fourier transform to MR magnitude images [5–7]. This is because raw k-space data is typically not available from large population MRI data sets such as the UK Biobank and the German National Cohort (GNC) [8, 9]. The prevalence of motion artifacts can also vary between subject groups, for instance between healthy controls and patients suffering from neurodegenerative disorders [10–12]. This will adversely affect machine learning-based prediction models if not addressed [13], underscoring the need to generate images with synthetic motion artifacts to obtain unbiased data sets. However, systematic large-scale evaluations are needed to evaluate how motion-corrupted images, generated using magnitude-based simulations, compared to real motion-corrupted data.

In this work we explored whether a magnitude-based approach could yield realistic motion-corrupted images. To do this, we utilized the publicly available Movement-Related Artifacts (MR-ART) dataset made up of structural magnitude data with and without real subject motion [14]. We simulated the two described motion paradigms on the motion-free images and compared the results to the corresponding real motion-corrupted images. We performed these simulations using the functionality available in the open-source Python framework TorchIO [15]. TorchIO facilitates the standardization of medical image data augmentation for deep learning projects. Included in its library are standard augmentation techniques, such as flipping and spatial transformations, as well as MRI-specific modifications, such as those for bias field and motion. Augmentation functions are typically random, generating different results each time they are called [16]. The MRI-specific simulations in TorchIO follow this convention and are thus not designed to simulate specific motion paradigms and/or pulse sequences. This "random motion" approach is not unique to TorchIO and has been implemented in other studies [6, 17, 18]. Here, we modified the original motion simulation functionality in TorchIO to better mimic the specific motion paradigm and MR pulse sequence parameters of the real motion-corrupted data in the MR-ART dataset. To gauge differences between the real and simulated (both original and modified) motion-corrupted images, we calculated image quality metrics (IQMs). We evaluated the outcome of the simulated motion corruption on brain cortical segmentation using the Dice-Sørensen coefficient, which also provided a measure of the robustness to motion of six publicly available whole brain segmentation tools.

## Methods

### The MR-ART dataset

Simulations were performed on the publicly available MR-ART dataset consisting of 3T MPRAGE defaced NIfTI image volumes at 1 mm$^3$ isotropic resolution of 148 subjects [14]. The authors of the dataset reported that all participants provided written, informed consent

**Table 1. Summary of acquired and simulated image volumes.**

| Name | Description | Pitch/Yaw/Roll [°] | #Rigid Transforms | #Nods |
|---|---|---|---|---|
| NoMotion | Motion-free baseline image | N/A | N/A | N/A |
| Real5 | 5 actual nods performed during acquisition | ?/0/0 | N/A | 5 |
| Real10 | 10 actual nods performed during acquisition | ?/0/0 | N/A | 10 |
| Ori5 | 5 transforms with original motion simulation | 0-15/0-15/0-15 | 5 | N/A |
| Ori10 | 10 transforms with original motion simulation | 0-15/0-15/0-15 | 10 | N/A |
| Mod5 | 5 nods with modified motion simulation | 15/0/0 | 20 | 5 |
| Mod10 | 10 nods with modified motion simulation | 15/0/0 | 40 | 10 |

before participation, and that the study was approved by the National Institute of Pharmacy and Nutrition (file number: OGYÉI/70184/2017). Since only secondary analysis of human data was performed here, a separate ethical approval was not obtained. The data was accessed on the 27th of March 2023. All data had been pseudoanonymized and defaced using PyDeface. No other information was available that could be used to identify individual participants. The images were acquired on a Siemens Magnetom Prisma 3T system (Siemens Healthcare GmbH, Erlangen, Germany) with a 20-channel head-neck receive coil. In this dataset, three images were acquired per subject. During the scans, each subject was instructed via a visual cue to: (1) stay still, (2) nod 5 times, and (3) nod 10 times. The instructions to nod were evenly distributed across the acquisition. A 'nod' was defined as tilting the head up along the sagittal plane (a pitch rotation) and then returning to the original position. The rotation in degrees and the duration of the nod was not given although it was noted that the visual cue was presented for 5 s. From this point on, the motion-free image volume is referred to as 'NoMotion' while the images acquired under 5/10 nods are referred to as 'Real5'/'Real10' respectively (STAND/HM1/HM2 in the original publication) (Table 1).

## Motion simulations

TorchIO version 0.18.73 was used as the basis for the motion simulations [15]. The MR-ART NIfTI files were loaded into TorchIO as a 'SubjectsDataset' consisting of 148 'Subject' objects which each contained a single image volume represented by an image class. From here, additional preprocessing and augmentation can be performed. Detailed information about the general TorchIO data processing can be found here: https://torchio.readthedocs.io/index.html. In this work, the 'RandomMotion' function from the TorchIO library was applied to the NoMotion NIfTI volume of all 148 subjects. The function takes two floating point ranges for rotation (°) and translation (mm) along with one integer for the number of discrete movements. This implementation allows a wide range of different motion artifacts to be generated but is ill-suited for simulating a specific motion paradigm. Given these constraints, only a very rough simulation of the motion paradigm of the MR-ART study could be performed. A movement with a rotational range of 0–15° in either of the three axes (pitch, yaw, and roll) and no translational range (0–0 mm) was simulated either 5 or 10 times. The rotation/translation is by design randomly distributed across the three axes. This means that each axis will be assigned different values within the same range. For example, 5 simulated movements with a rotational range of 0–15° could result in a pitch = 4°/13°/15°/8°/5°, a yaw = 13°/0°/6°/0°/5°, and a roll = 0°/2°/7°/4°/11°. New numbers are generated each time the 'RandomMotion' function is applied, leading to a wide range of generated motion artifacts in between subjects in the generated dataset. The motion simulation itself was performed by rigidly (6 degrees of freedom) transforming the image volume to mimic different positions of the subject [17]. A Fourier transform was applied to each rigidly- transformed image. The resulting 'pseudo' k-spaces were combined

based on the timings of the movements into a 'composite' k-space. An inverse Fourier transform was applied to the composite k-space to yield the simulated motion-corrupted image. The periodicity of the movements is, by design, randomly distributed throughout the acquisition and was not evenly spaced as was the case in the MR-ART study. Further, since the phase encoding direction is not a parameter in the 'RandomMotion' function, it was not possible to directly control the direction of the ghosting/ringing artifacts. Instead, the direction of the artifacts depended on the orientation of the NIfTI volume when loaded into Python. The image volumes obtained by this original simulation are denoted 'Ori5/Ori10' respectively (Table 1).

To simulate the nodding motion of the MR-ART study more correctly, the 'RandomMotion' function was modified to allow a well-defined motion paradigm, where rotation and translation could be varied independently across the axis (e.g., the pitch could be defined independently of the yaw). It also allowed non-random timings of the movements and to expressly define the filling order of the 'pseudo k-space' so that ghosting/ringing artifacts appeared in the phase encoding direction. With these modifications in place, each individual nod could be defined as 4 rigid transforms to capture intermediate head positions along the full nodding motion. Either 5 or 10 nods were simulated, involving 20 or 40 transforms respectively and evenly distributed throughout the acquisition. The pitch magnitude was empirically set to 15˚, while a nod duration of 2.5 s was deemed reasonable as it was assumed that each subject performed a nod within 0–5 s, based on a 5 s visual cue. The simulated acquisition duration and phase encoding direction was the same as the actual ones, i.e., 316 s (5:16 min) and anterior-posterior. The image volumes obtained by this modified simulation are denoted 'Mod5/Mod10' respectively (Table 1). Following the Ori5/Ori10 and Mod5/Mod10 simulations, we had a total of 7 groups in the dataset. Therefore, the simulations yielded an augmented dataset of 148×7–8 = 1028 image volumes (8 image volumes were missing in the original dataset: 7 from Real5 and 1 from Real10). Fig 1 shows a schematic of the modified motion simulation.

## Image quality

To perform a quantitative comparison of the image quality between real and simulated motion-corrupted images we used the MRI Quality Control tool (MRIQC) [19]. MRIQC allows for automatic extraction of an array of image quality metrics (IQMs) for objective quality control of MR neuroimaging data. A number of these IQMs have been reported to correlate with head motion, namely the Coefficient of Joint Variation (CJV) [20], the Entropy Focus Criterion (EFC) [21], and the quality indices QI1/QI2 [22]. However, both EFC and QI1/QI2 rely on the assumption that MRI artifacts result in increased signal intensity in the image background and are thus not appropriate for image data that has undergone filtering and/or masking such as defacing. Here we focused on MRIQC-based IQMs that barely use background pixels in their calculation: CJV, SNR, and CNR. The EFC and QI1/QI2 are included as supporting information.

Before MRIQC processing, a defaced mask was created for each subject, based on the NoMotion image, and applied to the motion simulated images. This removed pixels that were not present in the original masked dataset and thus facilitated a more correct comparison. Thereafter, all 148x7-8 = 1028 images were run through the MRIQC pipeline. Processing was performed using an MRIQC Docker container, version 23.1.0.

**Coefficient of Joint Variation (CJV).** The CJV incorporates information about the intensity distribution within, and the contrast between, segmented white matter (WM) and gray matter (GM). It is calculated as:

$$\text{CJV} = (\sigma_{\text{WM}} + \sigma_{\text{GM}})/(|\mu_{\text{WM}} - \mu_{\text{GM}}|) \tag{1}$$

where σ denotes the standard deviation and the mean of the respective segmented tissue

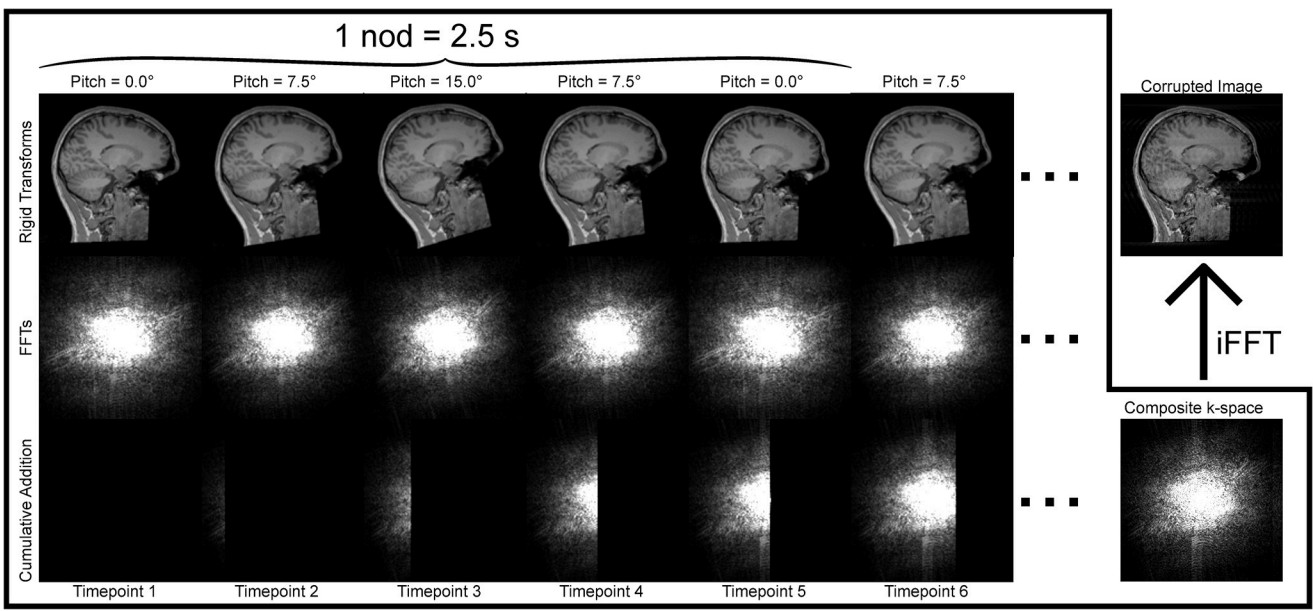

**Fig 1. Schematic of the modified motion simulations of a nodding motion paradigm.** Rigid transformations are applied to the baseline NoMotion image volume (top left corner), rotating the image around the right-left axis in a pitch rotation (top row). The pitch is applied with either 7.5˚ or 15.0˚ where 7.5˚ designates the 'halfway' point of the nod. FFTs are applied to each of the images to obtain "pseudo" k-spaces (middle row). These k-spaces are masked based on which time point the subjects head was in the corresponding position and then cumulatively added (bottom row) to eventually form a composite k-space (bottom right corner). Lastly, an iFFT is applied to the composite k-space, resulting in the motion-corrupted image (top right corner). The absolute value of the complex-valued k-spaces are displayed here. The simulated scan duration was reduced for illustrative purposes.

classes. Higher values are associated with poor image quality and motion artifacts. It was originally used to optimize an intensity non-uniformity correction algorithm [20].

**Signal-to-Noise-Ratio (SNR).** The SNR reported here is based on variance between different tissues, ignoring the air background. It is calculated as:

$$\text{SNR}_{\text{WM,GM,CSF}} = \mu_{\text{WM,GM,CSF}} / (\sigma_{\text{WM,GM,CSF}} \sqrt{n/(n-1)}) \tag{2}$$

where $n$ is the number of pixels in the respective tissue mask. The total SNR is then obtained simply as the mean across the three tissue classes:

$$\text{SNR} = (\text{SNR}_{\text{WM}} + \text{SNR}_{\text{GM}} + \text{SNR}_{\text{CSF}})/3. \tag{3}$$

**Contrast-to-Noise-Ratio (CNR).** The CNR refers to the contrast-to-noise-ratio between WM and GM. It is calculated as:

$$\text{CNR} = (|\mu_{\text{GM}} - \mu_{\text{WM}}|)/\sqrt{\sigma_{\text{air}}^2 + \sigma_{\text{WM}}^2 + \sigma_{\text{GM}}^2} \tag{4}$$

where $\sigma_{\text{air}}$ is the standard deviation of the air background. The CNR was included since the background pixel dependency was fairly small compared to EFC, and $\text{QI}_1/\text{QI}_2$.

## Automatic brain segmentation

Complementing the MRIQC comparison, automatic brain segmentation was performed to compare differences in segmentation performance between real and simulated motion-corrupted data. Whole brain segmentation of the cerebral cortex was performed using 6 freely available segmentation tools: FreeSurfer [23], BrainSuite [24], ANTs [25], SAMSEG [26], FastSurfer [27], and SynthSeg+ [28, 29]. FreeSurfer, SAMSEG, FastSurfer, and SynthSeg+ all use

the same labeling system for segmented brain structures, making comparison between segmentations of the cerebral cortex straightforward. The cerebellar cortex was excluded since the cortical segmentation in ANTs does not include it. As BrainSuite does not directly output a cortical segmentation, we derived a NIfTI volume that could be compared to the output of the other segmentation tools by combining three masks: A mask of the boundary between white matter and cortical gray matter, a mask of the total gray matter (cortical and deep gray matter) based on a gray matter probability map (pixels with >50% probability of belonging to gray matter was kept), and a mask of the cerebrum. This analysis doubled as a comparison of the motion robustness of the range of segmentation software.

Segmentations of real motion-corrupted images were rigidly coregistered to the segmentation of the baseline NoMotion image using FSL Flirt, nearest neighbor interpolation, and transformation matrices obtained from coregistering the corresponding magnitude images. The aseg.mgz output from FreeSurfer, the aparc.DKTtlas+aseg.deep.mgz output from FastSurfer, and the seg.mgz output from SAMSEG was converted to native space using FreeSurfer mri_label2vol and converted from.mgz format to compressed NIfTI format using dicm2nii in MATLAB. FreeSurfer was run using v6.0. while SAMSEG and SynthSeg+ were run using v7.3 and v7-dev respectively. ANTs was run on v.2.3. BrainSuite and FastSurfer were both run through Docker images, the former on v21a and the latter on v2.0.

**Data analysis.** The IQM = {CJV, CNR, SNR} and the Dice-Sørensen Coefficient (DSC) were used to quantitatively analyze the decrease in image quality and segmentation performance imposed by subject motion. It was explored whether Mod = {Mod5, Mod10} could yield results closer to Real = {Real5, Real10} compared to Ori = {Ori5, Ori10}. The DSC was calculated as

$$
\text{DSC} = \frac{2|\text{Seg}(\text{NoMotion}) \cap \text{Seg}(\text{Motion})|}{|\text{Seg}(\text{NoMotion})| + |\text{Seg}(\text{Motion})|}
\tag{5}
$$

where Seg(NoMotion) and Seg(Motion)∈{Seg(Real5), Seg(Real10), Seg(Ori5), Seg(Ori10), Seg(Mod5), Seg(Mod10)} are the respective cortical segmentations. Comparisons were performed using a combination of scatter and probability density (raincloud) plots and box plots. For the MRIQC analysis, linear least squares fits were calculated between the IQMs of the motion simulated images, IQM(Sim) = {IQM(Ori), IQM(Mod)}, vs. the IQMs of the real motion-corrupted images, IQM(Real). The values of these linear functions were calculated and compared between Ori = {Ori5, Ori10} and Mod = {Mod5, Mod10} where $r^2 = 1$ would indicate perfect agreement with Real = {Real5, Real10}. Lastly, Bland-Altman plots of IQM(Sim)–IQM(Real) vs. (IQM(Sim) + IQM(Real))/2 where the mean of the former was compared between Ori and Mod. To ensure that Seg(NoMotion) was reasonably comparable across segmentation tools, another DSC was calculated as

$$
\text{DSC} = \frac{2|\text{Seg}_{\text{FS}}(\text{NoMotion}) \cap \text{Seg}_{\text{other}}(\text{NoMotion})|}{|\text{Seg}_{\text{FS}}(\text{NoMotion})| + |\text{Seg}_{\text{other}}(\text{NoMotion})|}
\tag{6}
$$

where $\text{Seg}_{\text{FS}}(\text{NoMotion})$ denotes the FreeSurfer segmentation and $\text{Seg}_{\text{other}}(\text{NoMotion})$ denotes the segmentation of any of the other 5 segmentation tools. FreeSurfer was used as a silver standard reference because of its most widespread use. All analysis was performed using R version 4.2.1 and MATLAB R2021a. The entire process, from simulations to data analysis, is outlined in Fig 2.

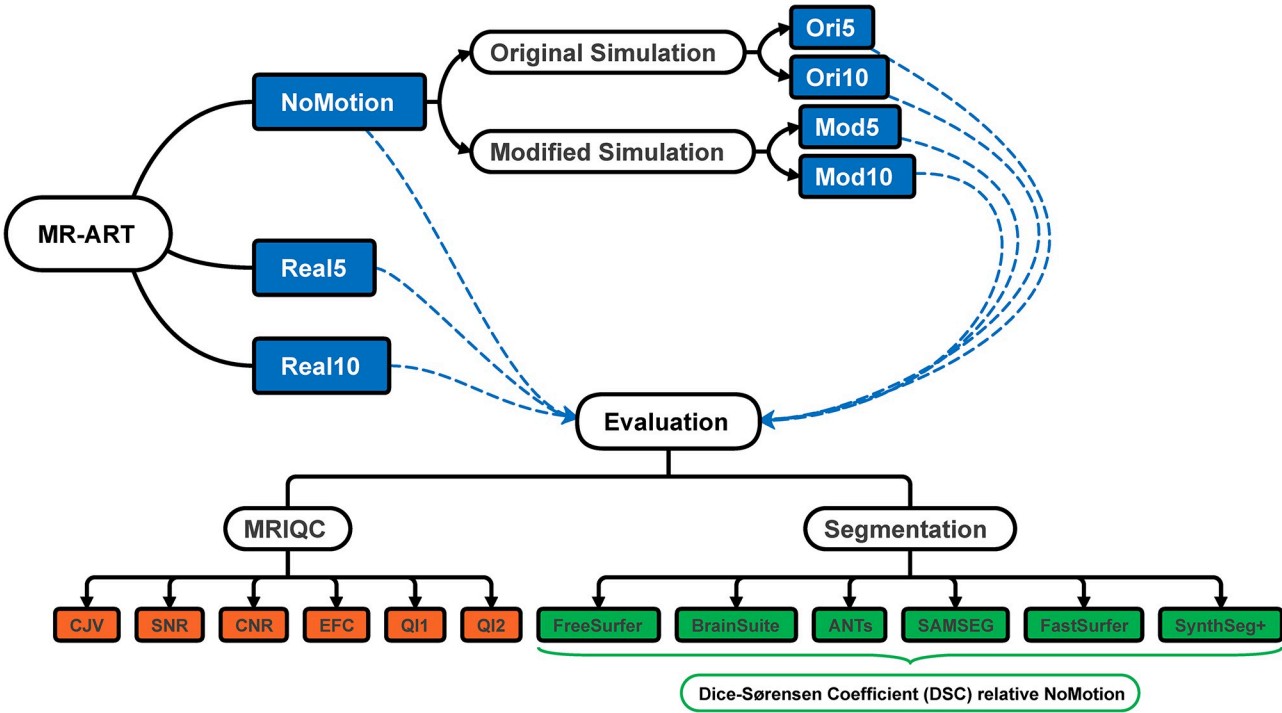

**Fig 2. Flowchart visualization of entire experimental procedure (see Methods for full description).** Subsets of the main MR-ART dataset are denoted in blue, IQMs in orange, and segmentation tools in green.

## Results

### Qualitative characteristics of real and simulated motion artifacts

Distinct ringing artifacts were visible in the real motion-corrupted images (Real5, Real10), especially in cortical areas. When performing the simulations (Fig 1), these ringing artifacts were mimicked closer to the real motion-corrupted images with the modified simulations (Mod5, Mod10). Images created with the original simulations (Ori5, Ori10) did not display these ringing artifacts as clearly but instead appeared as more blurred and with a worse overall image quality. Two representative subjects are shown in the axial (Fig 3) and sagittal (Fig 4) projections. Note that the exact motion paradigm is not known on an individual level for Real5/Real10 and Ori5/Ori10.

### Outcome on image quality

In the real motion-corrupted images (Real5, Real10), we observed a clear trend of decreasing image quality with increasing nodding frequency (Fig 5). This was also the case for the modified simulations (Mod5, Mod10). In contrast, it was not possible to induce a monotonic decrease in image quality using the original simulations (Ori5, Ori10). The modified motion simulations generally resulted in images with IQM distributions closer to those of the real motion-corrupted data compared to those generated with the original simulation. We observed a larger spread in the IQM distribution of the real motion-corrupted data compared to both the original and modified simulations. This was also visible upon inspection of the MR-ART dataset; the degree of motion corruption within both Real5 and Real10 varied substantially amongst subjects. As expected, the spread of the IQM distribution was reduced using the modified simulations. Surprisingly, the IQM spread was also reduced for the original

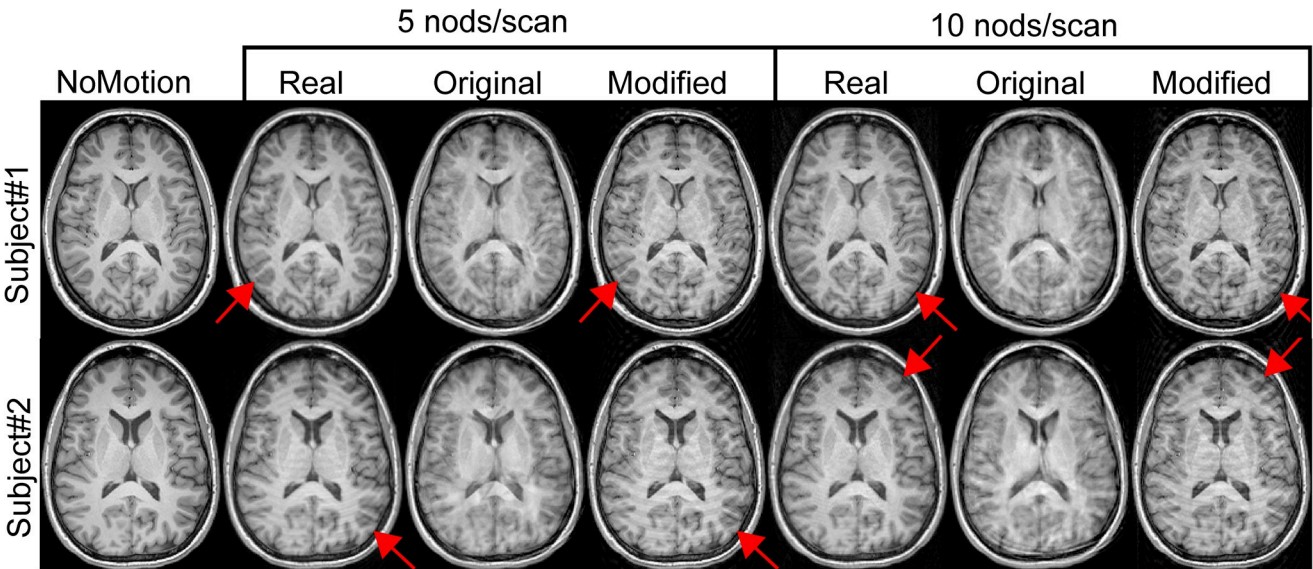

**Fig 3. Motion simulation results of two subjects using either the original TorchIO implementation or our modified version compared to images acquired under real motion.** Real/simulated motion was performed/simulated at either 5 or 10 nods per scan. The leftmost column shows the motion-free baseline image. The ringing artifacts in cortical areas (arrows), characteristic of the Real images, are better represented in the modified than the original simulations.

simulations, despite their random character. Similar to the original study [14], the EFC showed a correlation with the motion level (S1 Fig), albeit weaker than the primary IQMs. The IQMs $QI_1$ and $QI_2$ did not show conclusive results across the nodding frequencies, also for the real motion-corrupted data (S2 and S3 Figs).

The linear regression analysis (Fig 6, upper panel) and Bland-Altman plots (Fig 7, lower panel) revealed improved agreement between simulations and real data with our modified simulations. The $r^2$ increased from 0.017/0.034/0.055 to 0.23/0.24/0.23 for CJV/CNR/SNR respectively where $r^2 = 1$ would indicate perfect agreement with Real = {Real5, Real10}. In the Bland-Altman plot, the mean difference to the Real = {Real5, Real10} reference decreased from

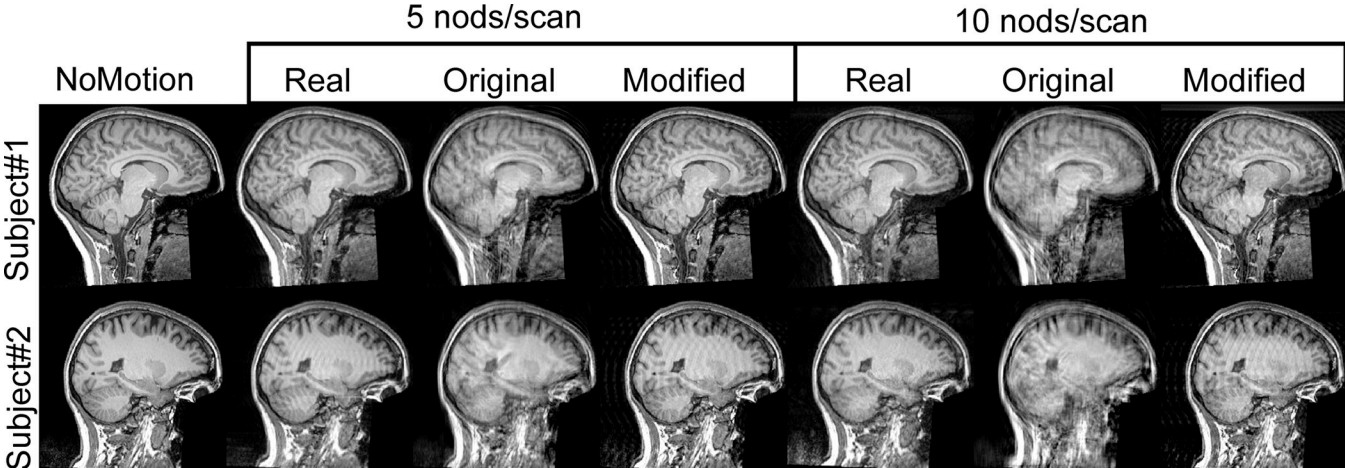

**Fig 4. Simulated and real motion-corrupted images in the sagittal plane showing the defacing.** See the corresponding axial representation in Fig 2 for a description.

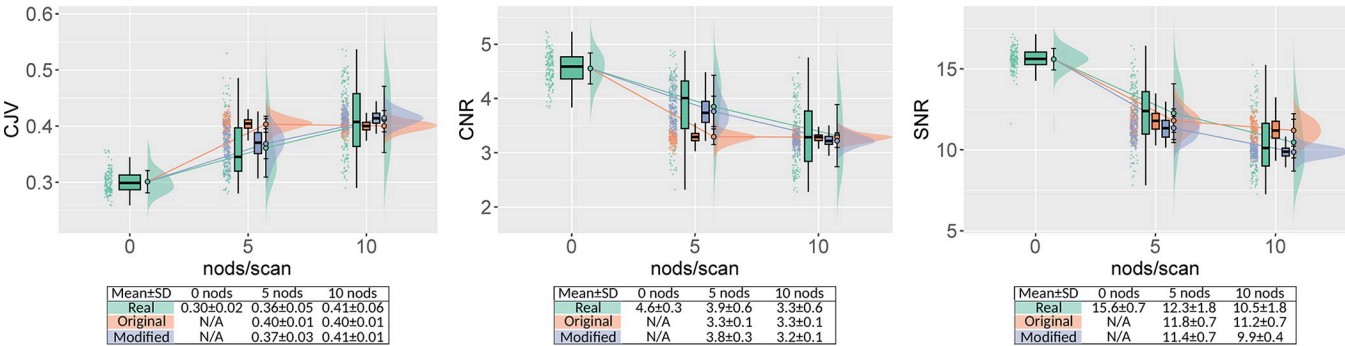

**Fig 5. MRIQC-calculated background-independent IQMs for real motion (green), the original simulated motion (orange), and the modified simulated motion (blue) for 148 subjects and across three nodding frequencies: The NoMotion baseline (0 nods/scan), 5 nods/scan, and 10 nods/scan.** Results are visualized as probability density functions with corresponding jittered scatter plots (raincloud plots), and box plots. Means and standard deviations are denoted in the legend and by filled circles with connecting lines between motion levels. From left to right: Coefficient of Joint Variation (CJV), Contrast-to-Noise-Ratio (CNR), and Signal-to-Noise-Ratio (SNR). Overall, there is an improved agreement between real and simulated motion for the modified version. The monotonic dependency on nodding frequency is much clearer for the modified version. Note also the much larger spread in the real motion data.

0.015/-0.29 to 0.004/-0.086 for CJV/CNR but increased from 0.17 to -0.73 for SNR. The separation between Mod5 and Mod10 was clearly visible, which was not the case for Ori5 and Ori10.

## Outcome on automatic brain segmentation

Motivated by the prevalence of ringing artifacts in cortical areas (Figs 3 and 4), the cerebral cortex was segmented and compared between uncorrupted images and those corrupted with real or synthetic motion. Fig 7 shows a representative example of the detrimental effect of subject motion, captured by the DSC, across the 6 segmentation tools. The decrease in DSC differed amongst the segmentation tools, where Synthseg+ showed the smallest decrease. However, upon qualitative observation Synthseg+ appeared to generate thicker/smoother cortical segmentations which extended into the sulci in some areas (Fig 7). For motion-free images, FreeSurfer, FastSurfer, and SAMSEG yielded the most high-resolution segmentation,

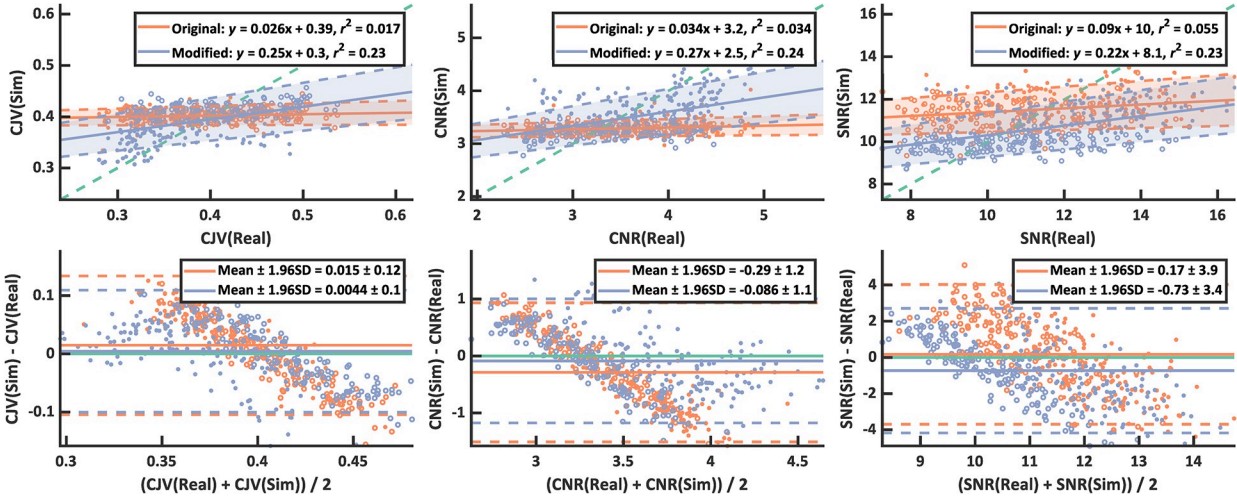

**Fig 6.** Linear regression plots (top row) and Bland-Altman plots (bottom row) of the CJV (left), CNR (middle), and SNR (right) showing the improved agreement between real (Real) and simulated (Sim) motion using the modified (blue) compared to the original (orange) motion simulation. The identity line/zero line in the regression/Bland-Altman plots are denoted in green. With the modified simulation, the r² of the linear fit is higher and the separation between IQMs from images with 5 nods (filled datapoints) and 10 nods (empty datapoints) is clear.

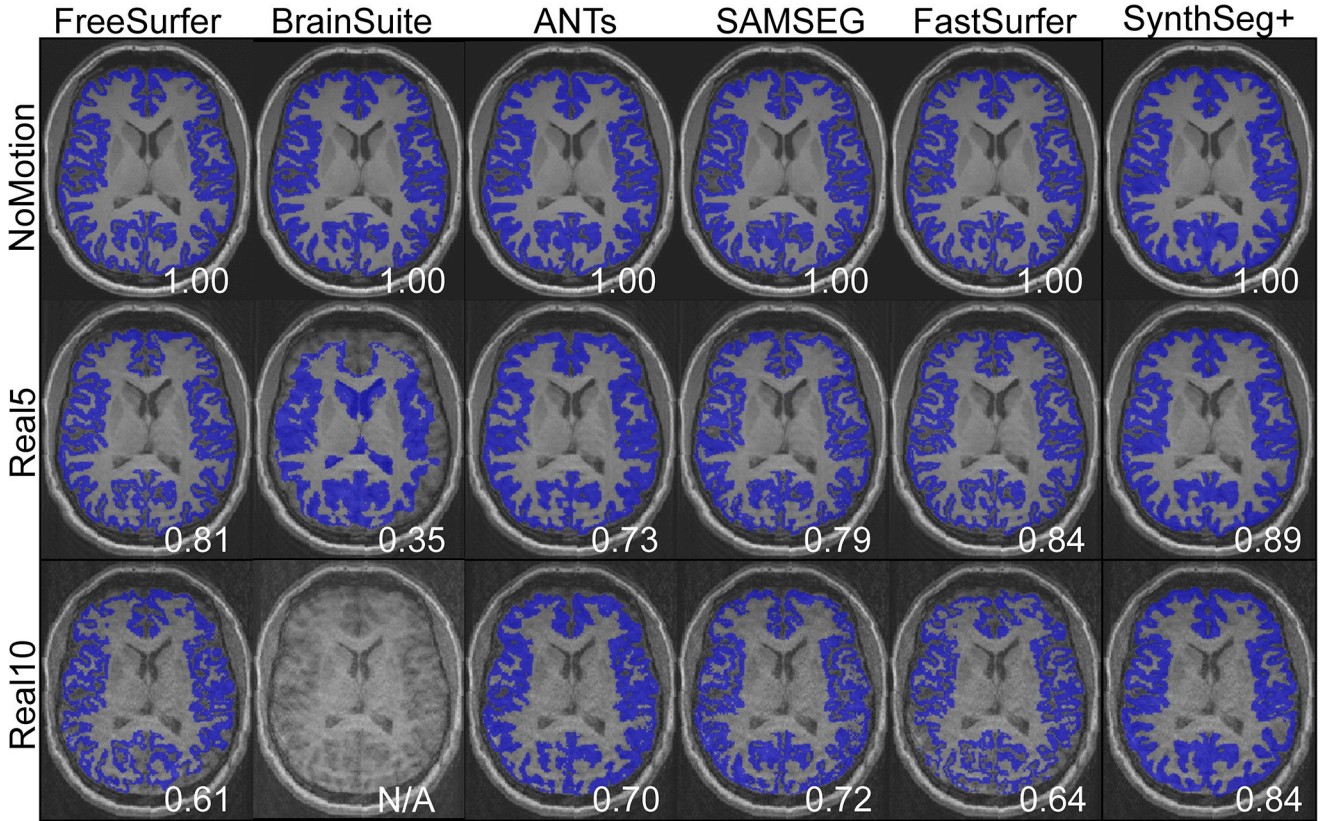

**Fig 7. Effect of motion on brain segmentation tools for an example subject.** Rows denote real motion at the three different motion levels: None (NoMotion), 5 nods per scan (Real5), and 10 nods per scan (Real10). Columns denote the six segmentation tools (FreeSurfer, BrainSuite, ANTs, SAMSEG, FastSurfer, and SynthSeg+). For each segmentation, the DSC relative the NoMotion segmentation for that tool is shown in the bottom right corner. Note the higher DSCs of the contrast-agnostic SynthSeg+. SynthSeg+ further stands out in that the cortical segmentation appears thicker and more 'smooth'. BrainSuite did not yield any output for Real10.

followed by ANTs. BrainSuite performed rather poorly, and a majority of the segmentations applied to the real motion-corrupted data (Real5, Real10) failed to process.

Fig 8 shows the decrease in DSC relative to the motion-free reference (NoMotion) for Real = {Real5, Real10}, Ori = {Ori5, Ori10}, and Mod = {Mod5, Mod10} for all 6 segmentation tools. The average differences in the DSC across the segmentation tools between Real = {Real5, Real10} and Ori = {Ori5, Ori10} were -0.13±0.08 (5 nods/scan) and -0.19±0.08 (10 nods/scan). When using our modified simulation method, we observed improved agreement with the real motion-corrupted data, also with respect to segmentation performance. The average differences in DSC were reduced to +0.05±0.04 (5 nods/scan) and +0.02±0.05 (10 nods/scan). BrainSuite was excluded when calculating these average differences due to the poor performance already apparent on the real motion-corrupted data (Figs 7 and 8). SynthSeg+ had the weakest correlation to the real/simulated nodding frequency with a Pearson correlation coefficient of $r = -0.37$ compared to $r = -0.79/-0.85/-0.51/-0.54/-0.51$ for FreeSurfer/BrainSuite/ANTs/SAMSEG/FastSurfer respectively. This indicates a higher robustness to motion compared to the other methods.

We also compared the different segmentation tools with the silver standard FreeSurfer. For the NoMotion baseline images, FreeSurfer clearly showed the highest agreement with FastSurfer with a mean DSC = 0.95±0.03 (Fig 9). The remaining segmentation tools showed comparatively very similar DSCs.

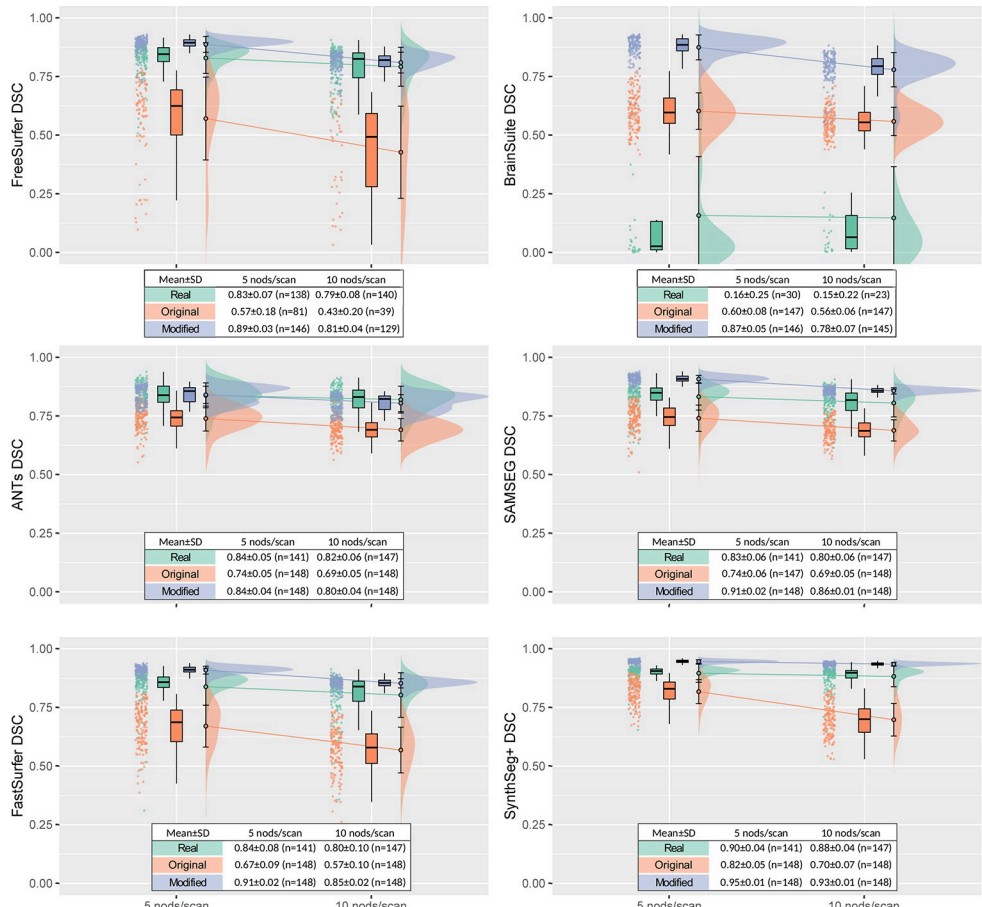

**Fig 8. Segmentation tool performance on real and simulated motion-corrupted data.** The DSC of the cerebral cortical segmentation obtained from motion-corrupted data relative the baseline NoMotion segmentation output was calculated for 6 segmentation tools: FreeSurfer, BrainSuite, ANTs, SAMSEG, FastSurfer, and SynthSeg+. Real motion-corrupted data (green), the original motion simulation (orange), and the modified simulation (blue) is shown across two levels of motion: 5 nods per scan, and 10 nods per scan. DSC = 1 for NoMotion. The number of calculated DSCs are shown in the legend since the segmentation pipeline did not always complete successfully when applied to the motion-corrupted data. The decrease in DSC at an increasing number of nods observed for the real data was more accurately mimicked by the modified compared to the original simulations. SynthSeg+ showed the smallest decrease in DSC of all tools.

## Discussion

We induced synthetic motion artifacts on magnitude MRI data and performed large-scale quantitative comparisons to real motion-corrupted data. Our purpose was to validate the use of magnitude-based motion simulations. To this end, we used the MR-ART dataset as a ground truth reference. To accurately simulate the motion paradigm performed in MR-ART, the random elements in the motion simulation of an existing data augmentation framework (TorchIO) needed to be removed. We made three modifications to the original motion simulation, explicitly specifying: (1) the rotation along each axis independently of each other, (2) the timings and durations of each discrete movement, and (3) the direction of the inner phase encoding direction. After implementing these simple changes, the real motion artifacts were more accurately mimicked in terms of artifact characteristics, image quality, and segmentation performance. It should be noted that the random characteristic of the original motion simulations in TorchIO is not an inherent weakness of the framework but rather an effective feature

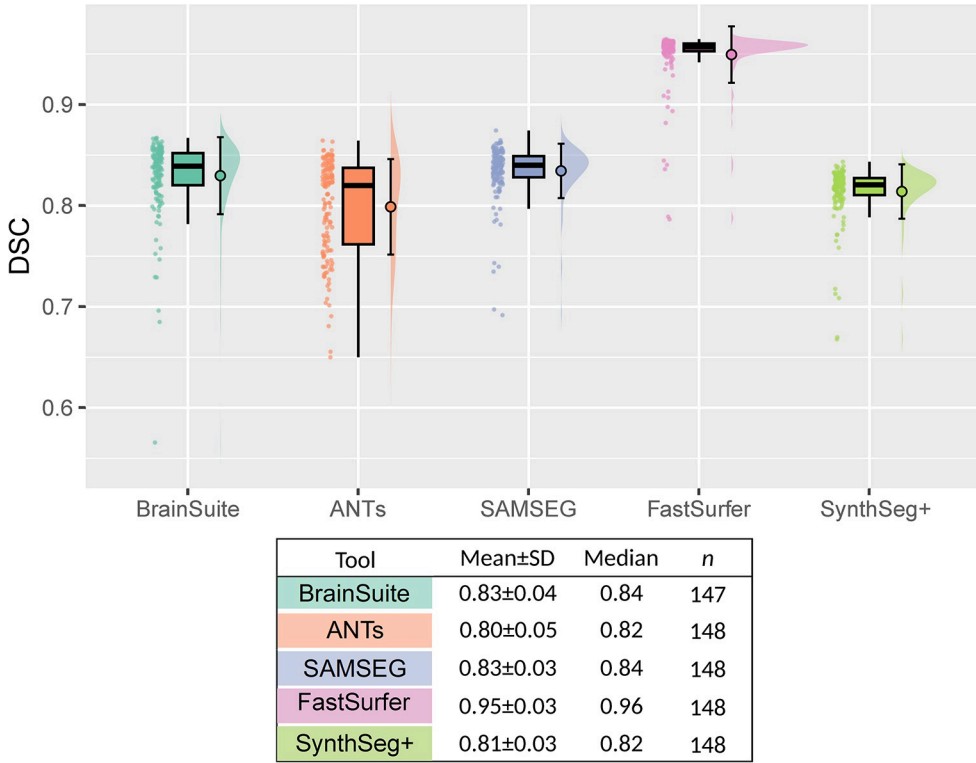

| Tool | Mean±SD | Median | n |
|---|---|---|---|
| BrainSuite | 0.83±0.04 | 0.84 | 147 |
| ANTs | 0.80±0.05 | 0.82 | 148 |
| SAMSEG | 0.83±0.03 | 0.84 | 148 |
| FastSurfer | 0.95±0.03 | 0.96 | 148 |
| SynthSeg+ | 0.81±0.03 | 0.82 | 148 |

**Fig 9. Comparison to FreeSurfer on baseline no-motion data.** The DSC of the cerebral cortex relative the FreeSurfer segmentation was calculated for the remaining 5 tools on the NoMotion images for all 148 subjects. FastSurfer produced the most FreeSurfer-like segmentation. The processing crashed for one of the subjects when using BrainSuite, hence n = 147.

to artificially increase training data variability for deep learning applications. However, when a specific motion paradigm and/or pulse sequence is to be simulated, modifications as the ones described here become necessary.

All primary IQMs—CJV, CNR, and SNR—indicated a better agreement in image quality between the real motion-corrupted images and images corrupted with our modified motion simulations compared to the original simulations. This is because the randomness of the elements in the original motion simulations (including the phase encoding direction) were removed. The MR-ART dataset does not contain random motion [14], thus the original random features of TorchIO did not accurately simulate the motion paradigm. The modified simulations were especially more effective in capturing the progressive worsening in image quality with increasing nodding frequency, compared to the original simulations. The improved agreement was also visible on a qualitative level, where the characteristic ringing artifacts of the MR-ART dataset could be much better simulated using the modified simulations. Here, the random behavior of the original simulations instead results in blurring and perceived overall degraded image quality. The IQMs $QI_1$ and $QI_2$ showed inconclusive results across different (even real) nodding frequencies. It should be noted that the data was defaced making the air background-based analysis not possible [22]. On the other hand, the EFC estimates were very similar to those published by Nárai et al. [14] in the original MR-ART publication, indicating that the defacing did not have a particularly large impact on that metric.

We observed a deterioration in segmentation performance, similar to the changes seen in the primary IQMs. The DSCs for the modified simulations were much closer to the DSCs of

the real motion-corrupted data compared to the DSCs of the original simulations. The choice to focus on the cortical segmentation was motivated by visual inspection of the MR-ART data which showed pronounced ringing in cortical areas. This choice was also supported by the recognition that head motion strongly affects cortical gray matter volume and thickness estimates [30, 31].

Our segmentation-based analysis doubled as an evaluation of the robustness of 6 popular brain segmentation tools. Here, SynthSeg+ stood out in terms of relatively small changes in the DSC with increasing nodding frequency. This insensitivity to motion could in part be due to a generally thicker cortical segmentation with a 'smoothed' appearance that partly extend into the sulci. This hypothesis is supported by previous work which reported higher total GM volumes using SynthSeg (albeit not SynthSeg+) compared to FreeSurfer, SAMSEG, and FastSurfer [32]. However, SynthSeg+ did not show a markedly worse agreement with the silver standard FreeSurfer in the baseline motion-free images compared to most other segmentation tools. FastSurfer did not show a much higher DSC across nodding frequencies compared to FreeSurfer, as was previously reported [33]. Also considering the performance of ANTs and SAMSEG, we did not find clear support for a general motion robustness increase in CNN-based methods (SynthSeg+, FastSurfer). van Nederpelt et al. [32] reported markedly lower intra-class correlation coefficients on data from repeated measurements when using FreeSurfer, compared to SAMSEG, FastSurfer, and SynthSeg (albeit not SynthSeg+). This could be related to the motion robustness analysis performed in our study, although here it is SynthSeg + that stands out in terms of high DSCs. The poor performance of BrainSuite, observed in this work, could possibly be explained by the fact that no direct cortical segmentation is available from the output, although this does not explain why processing failed for most of the real motion-corrupted data.

The modified motion simulation described here has been applied to data from the ADNI (Alzheimer's Disease Neuroimaging Initiative, [34]) to mitigate biases in the distribution of motion artifacts between healthy controls, subjects with mild cognitive impairment, and AD patients. The aim of this ongoing work is to disentangle structural changes due to AD from motion artifacts and thus improve predictive performance.

The magnitude-based simulations performed here will not be as realistic as those performed on multi-channel k-space data [35], which are commonly not available for population or large-scale clinical MRI studies. Multi-channel phased arrays are typically employed for parallel imaging [36]. In the MR-ART dataset, a GRAPPA factor of 2 was applied, which means that every other k-space line is not measured but synthesized using weighting factors based on the measured data and indirect sensitivity measurements [37]. The interplay between the timing of discrete movements and the calculation of these weighting factors cannot be simulated using magnitude data alone. This is especially true when considering that receive sensitives are highly variable and depend on the position of the subject. Further, we have here focused only on simulating rotational motion where a rotation in image space results in an identical rotation in k-space. Translational motion will instead result in a linear phase ramp according to the Fourier shift theorem [38]. Although our suggested approach does not directly manipulate the pseudo k-space, it would be interesting to examine how a magnitude-based approach would compare to real-world data acquired when using translational motion instead of rotational motion.

Nevertheless, we show that the modified motion simulations compare well to real motion-corrupted data and are a very good alternative when only magnitude image data is available. We show here that it is essential to consider the type of motion paradigm and pulse sequence (phase encoding direction, acquisition duration) prior to embarking on motion simulation. This is especially true if multiple levels of artifact severity are needed, since we show that

simply increasing the number of transforms in the original TorchIO simulations was insufficient to induce a monotonic relationship between image quality and nodding frequency. One limitation of this work is the heterogeneity of the MR-ART dataset amongst different subjects. Based on visual inspection of the data as well as the IQM data distribution, it was evident that the same nodding frequency resulted in a large variation in artifact severity between subjects. The large IQM distribution for the real motion data is indicative of individual variations in the motion paradigm (pitch/duration) between subjects. In some cases, the lower nodding frequency resulted in a higher artifact severity. Kemenczky et al. [33] used radiologist image quality rating scores, available in the published dataset, instead of IQMs to evaluate the motion robustness of deep learning-based brain segmentation tools compared to FreeSurfer. Since we were interested in trying to emulate a specific motion paradigm, we instead opted for splitting the data based on the nodding frequency. Based on the monotonic relationship in both the IQMs and the DSCs with nodding frequency, we believe that the relatively large sample size of 148 subjects was enough to overcome this large spread in image quality.

## Conclusion

Reasonably realistic motion artifacts can be induced on brain MRI by magnitude-based simulations when combined with knowledge of head movement and k-space sampling. We derive this conclusion based on a large-scale comparison of IQMs and cerebral cortex segmentation performance between simulated and real motion artifacts. SynthSeg+ showed the highest motion robustness of tested brain segmentation tools although this may, at least in part, be due to a thicker baseline segmentation. Future work could use this simulation approach to mitigate biases in the distribution of motion artifacts between study groups and to provide an unbiased foundation for the training of machine learning based models.

## Supporting information

**S1 Fig. The Entropy Focus Criterion (EFC) showed good agreement between real/modified.** There is a monotonic increase for Real/Modified but not for Original. However, the change in EFC between nodding frequencies is relatively small.
(DOCX)

**S2 Fig. The quality index, QI1, showed inconclusive results, likely because of its dependency on background pixels.**
(DOCX)

**S3 Fig. The quality index, QI2, showed unreasonable results, likely because of its dependency on background pixels.**
(DOCX)

## Author Contributions

**Conceptualization:** Hampus Olsson, Jason Michael Millward, Ludger Starke, Sonia Waiczies.

**Data curation:** Hampus Olsson.

**Formal analysis:** Hampus Olsson, Jason Michael Millward, Thomas Gladytz, Sonia Waiczies.

**Funding acquisition:** Christoph Lippert, Sonia Waiczies.

**Investigation:** Hampus Olsson.

**Methodology:** Hampus Olsson, Jason Michael Millward, Ludger Starke, Tobias Klein, Jana Fehr, Wei-Chang Lai, Sonia Waiczies.

**Project administration:** Jana Fehr, Christoph Lippert, Sonia Waiczies.

**Resources:** Thoralf Niendorf.

**Software:** Hampus Olsson, Tobias Klein.

**Supervision:** Christoph Lippert, Thoralf Niendorf, Sonia Waiczies.

**Validation:** Hampus Olsson.

**Visualization:** Hampus Olsson.

**Writing – original draft:** Hampus Olsson, Sonia Waiczies.

**Writing – review & editing:** Hampus Olsson, Jason Michael Millward, Ludger Starke, Thomas Gladytz, Tobias Klein, Jana Fehr, Wei-Chang Lai, Christoph Lippert, Thoralf Niendorf, Sonia Waiczies.

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
