## [Decision Letter · Decision Letter 0]

2 Jan 2024

PONE-D-23-32888Simulating rigid head motion artifacts on brain magnitude MRI data – Outcome on image quality and segmentation of the cerebral cortexPLOS ONE

Dear Dr. Olsson,

Thank you for submitting your manuscript to PLOS ONE. After careful consideration, we feel that it has merit but does currently not fully meet PLOS ONE’s publication criteria as it currently stands. Therefore, we invite you to submit a revised version of the manuscript that addresses the points raised during the review process. 

We look forward to receiving your revised manuscript.

Kind regards,

Florian Ph.S Fischmeister

Academic Editor

PLOS ONE

“I have read the journal's policy and the authors of this manuscript have the following competing interests: HO is currently employed by Philips Healthcare. Remaining authors have declated that no competing interests exist.”

Reviewers' comments:

Reviewer's Responses to Questions

**Comments to the Author**

1. Is the manuscript technically sound, and do the data support the conclusions?

Reviewer #1: Yes

Reviewer #2: Yes

2. Has the statistical analysis been performed appropriately and rigorously? 

Reviewer #1: Yes

Reviewer #2: Yes

3. Have the authors made all data underlying the findings in their manuscript fully available?

Reviewer #1: Yes

Reviewer #2: Yes

4. Is the manuscript presented in an intelligible fashion and written in standard English?

Reviewer #1: Yes

Reviewer #2: Yes

5. Review Comments to the Author

Reviewer #1: The authors of this manuscript performed an extensive work in comparing real motion with "realistically" simulated motion.

The work highlights the need for careful design of the processing work to obtain reliable and reproducible final results, especially when external and more generalized software are adopted.

I only have some minor issues to raise:

1. More detailed information are needed regarding the processing of the DICOMs done via TorchIO.

2. The "RandomMotion" function generates motion randomly, as stated in the work. How was this addressed? Was the generation repeated for the same range of values to have more variability?

3. The motion corruption using RandomMotion looks worse than the original image. A comparison of the motion parameters should be performed and discussed.

4. How do your findings regarding the segmentation tools relate with the current literature? One source could be: https://discovery.ucl.ac.uk/id/eprint/10174895/.

Moreover:

1. In various parts of the manuscripts, references are reported only using the name of the first author (e.g. Kemenczky et al.), without reporting the actual reference number. References need to be harmonized and written to be easily searched by the reader e.g. either adding the reference number or the year of publication.

Reviewer #2: In this manuscript, the authors present a method for simulating rigid-body head motion artifacts in MRI data. They compare the artifacts with those from real head motion through image quality metrics (IQM) and segmentation results. The motivation for this work is to provide machine learning based models with augmented data. The motion simulation is in ‘pseudo’ k-space as the motion is applied to magnitude MRI data which is than Fourier transformed to produce k-space data. The k-space data from the various motions are combined into a composite k-space which is then inverse Fourier transformed to produce the motion corrupted MRI. The authors have tailored their simulation to more closely mimic real motion by modifying the TorchIO library in Python to independently control the rotation of each axis, the timing and duration of the motion, define the phase encode direction of the acquisition, and breaking down nodding motion into multiple positions, rather than 2.

The authors have compared their simulation method to the established motion simulation library in Python, Torch IO, and real motion corrupted data. The comparison is performed between the IQMs Coefficient of Joint Variation (CJV), SNR, and CNR. They also compared segmentation results from 6 different brain segmentation algorithms by calculating the Dice-Sorensen Coefficient (DSC). The author’s modified motion simulation method had better agreement to real motion corrupted data across all IQM and DSC measures than the original Torch IO method.

Overall, the authors have presented a useful motion simulation method for magnitude MRI data. They have also shown the importance of properly mimicking the MRI acquisition in the motion simulation to get more realistic results. Although it is a useful analysis, the results are not surprising. The results would be more meaningful if the authors could provide a justification for why the original TorchIO library was chosen for MRI motion simulation given its limitations.

Minor Concerns

1- There are several places in the manuscript where the authors speculate about other published work based on their findings (Methods>Image Quality 2nd paragraph, Discussion 2nd and 6th paragraphs.) The information should be rewritten to list the facts and remove speculation.

2- In the Conclusion, the authors state that their simulation of realistic motion artifacts helps to mitigate biases between study groups and studies. There is no data in the paper to support this claim and it should be removed. Hopefully, the authors can apply their method and show such data in the future.

3- Although the proposed simulation more closely matches the real data in IQM distributions, they are still quite different, as seen in Fig. 5. This is especially true as the motion increases from 5 to 10 movements. The authors discuss limitations of their simulated data to match the real motion data. There are additional limitations, in addition to not being multi-channel k-space data or subjects having variable movement. For example, real motion can occur during different parts of the pulse sequence: inversion or excitation RF pulses, or various times during the gradient switching for the readout. The motion can also occur during different steps in the phase-encode along two axes. The MR-ART data uses an acceleration factor of GRAPPA=2. This type of real data can also have an interaction between the GRAPPA kernel used in the reconstruction and the motion corrupted k-space data. The authors could add a discussion about some of the other factors that produce artifacts in real motion corrupted MRI data, if they could be simulated in k-space based data, and if they could be simulation with magnitude MRI data.

6. PLOS authors have the option to publish the peer review history of their article (what does this mean?). If published, this will include your full peer review and any attached files.

Reviewer #1: No

Reviewer #2: No

---

## [Author Response · Author response to Decision Letter 0]

6 Feb 2024

Response to Reviewers

Reviewer comments are in blue, replies in black and quotations from the manuscript are denoted by dashed lines.

Reviewer 1 (R1)

R1.1 - More detailed information is needed regarding the processing of the DICOMs done via TorchIO.

We have added more detailed information regarding the loading of data and the data structure within TorchIO. We’ve also included a reference to the TorchIO documentation for the convenience of the reader, in case a more extensive explanation is needed. Please note that no additional processing was performed on the NIfTI data other than the motion simulations described in the manuscript. Perhaps this point also related to R1.2 concerning the “RandomMotion” function. Please also see our response there. Methods->Motion simulations-> 1st paragraph now reads as:

The MR-ART NIfTI files were loaded into TorchIO as a ‘SubjectsDataset’ consisting of 148 ‘Subject’ objects which each contained a single image volume represented by an image class. From here, additional preprocessing and augmentation can be performed. Detailed information about the general TorchIO data processing can be found here: https://torchio.readthedocs.io/index.html.

R1.2 - The “RandomMotion” function generates motion randomly, as stated in the work. How was this addressed? Was the generation repeated for the same range of values to have more variability?

The RandomMotion function was not repeated to have more variability for the same subject. The function was applied twice on the motion-free image of each subject. The 1st time, 5 discrete motions were simulated (Ori5). The 2nd time, 10 discrete motions were simulated (Ori10). Each time, a unique discrete motion was simulated: a pitch, a yaw, and a roll. Each of these three variables were randomly selected from the defined range of 0-15°. The below table shows an example of what this could look like for a particular subject.

Ori5

Discrete movement #1 #2 #3 #4 #5

Pitch [°] 0 13 7 5 8

Yaw [°] 14 1 4 5 12

Roll [°] 7 14 15 10 14

Ori10

Discrete movement #1 #2 #3 #4 #5 #6 #7 #8 #9 #10

Pitch 7 0 10 3 13 15 3 7 4 12

Yaw 3 9 1 15 14 12 5 1 8 3

Roll 15 10 7 6 1 2 13 14 9 14

The discrete motions for both motion severity levels (5 and 10 discrete movements) were thus never the same for any subject but rather randomly different. This is now clearly stated in the manuscript together with a condensed version of the above example (Methods->Motion simulations->1st paragraph): 

For an example subject, 5 simulated movements with a rotational range of 0-15° could result in a pitch =4°/13°/15°/8°/5°, a yaw = 13°/0°/6°/0°/5°, and a roll = 0°/2°/7°/4°/11°. New numbers are generated each time the ‘RandomMotion’ function is applied, leading to a wide range of generated motion artifacts in between subjects in the generated dataset.

R1.3 - The Motion corruption using RandomMotion looks worse than the original image. A comparison of the motion parameters should be performed and discussed.

It is correct that Ori5/Ori10 displays an overall worse image quality compared to Real5/Real10 in Figures 3 and 4 as mentioned in Results->Qualitative characteristics of real and simulated motion artifacts. Unfortunately, a direct comparison of motion parameters in the individual case is difficult since the exact motion paradigm is not known in the MR-ART dataset (Methods->The MR-ART dataset and Discussion->2nd paragraph). It is also regrettably the case that motion simulation parameters on the individual level were not saved for Ori5/Ori10. This is now clearly stated (Results->Qualitative characteristics of real and simulated motion artifacts):

Note that the exact motion paradigm is not known on an individual level for Real5/Real10 and Ori5/Ori10.

We have also added a section in the Discussion reflecting on the qualitative differences that can be seen in the example subject (Discussion->2nd paragraph):

The improved agreement was also visible on a qualitative level, where the characteristic ringing artifacts of the MR-ART dataset could be much better simulated using the modified simulations. Here, the random behavior of the original simulations instead results in blurring and perceived overall degraded image quality.

R1.4 - How do your findings regarding the segmentation tools relate with the current literature? One source could be https://discovery.ucl.ac.uk/id/eprint/10174895/.

We thank the reviewer for this suggestion. We have included the reference to the study by Kemenczky P et al., Scientific reports 2022, in addition to our previous discussion (line 360) and now discuss the results by van Nederpelt et al., Neuroradiology 2023 in more detail (Discussion->4th paragraph):

This hypothesis is supported by previous work which reported higher total GM volumes using SynthSeg (albeit not SynthSeg+) compared to FreeSurfer, SAMSEG, and FastSurfer [32].

and

van Nederpelt et al. [32] reported markedly lower intra-class correlation coefficients on data from repeated measurements when using FreeSurfer, compared to SAMSEG, FastSurfer, and SynthSeg (albeit not SynthSeg+). This could be related to the motion robustness analysis performed in our study, although here it is SynthSeg+ that stands out in terms of high DSCs.

R1.5 - In various parts of the manuscripts, references are reported only using the name of the first author (e.g. Kemenczky et al.), without reporting the actual reference number. References need to be harmonized and written to be easily searched by the reader e.g. either adding the reference number or the year of the publication.

For better visibility and accessibility, the reference number is now included directly after the author’s name rather than at the end of the sentence.

 

Reviewer 2 (R2)

R2.1 - Although it is a useful analysis, the results are not surprising. The results would be more meaningful if the authors could provide a justification for why the original TorchIO library was chosen for MRI motion simulation given its limitations.

The “random motion” approach is not unique to TorchIO and has been implemented elsewhere. For instance, Shaw R, et al., IEEE transactions on medical imaging. 2020;39(9):2881-92 and Graham MS, et al., Neuroimage. 2016;125(15):1079-94. TorchIO is a user-friendly framework for standardization of medical image data pre-processing, which motivates using its in-built functionality for motion simulations as reference for machine learning purposes. Further, the random feature of the motion simulations in TorchIO follows the rationale for general image data augmentation, which is designed to artificially increase training data variability. It is thus warranted to investigate the impact of removing the random element in the motion simulations and to explore to what extent this makes the resulting images more comparable to real-life data. We agree that the justification for our approach was not made sufficiently clear in the original text. Thus, the following has been added to the Introduction (Introduction->3rd paragraph):

TorchIO facilitates the standardization of medical image data augmentation for deep learning projects. Included in its library are standard augmentation techniques, such as flipping and spatial transformations, as well as MRI-specific modifications, such as those for bias field and motion. Augmentation functions are typically random, generating different results each time they are called [16]. The MRI-specific simulations in TorchIO follow this convention and are thus not designed to simulate specific motion paradigms and/or pulse sequences. This “random motion” approach is not unique to TorchIO and has been implemented in other studies [6, 17-18].

and (Discussion->1st paragraph)

It should be noted that the random characteristic of the original motion simulations in TorchIO is not an inherent weakness of the framework but rather an effective feature to artificially increase training data variability for deep learning applications. However, when a specific motion paradigm and/or pulse sequence is to be simulated, modifications as the ones described here become necessary.

R2.2 - There are several places in the manuscript where the authors speculate about other published work based on their findings (Methods>Image Quality 2nd paragraph, Discussion 2nd and 6th paragraphs.) The information should be rewritten to list the facts and remove speculation.

The speculative statements have been removed. 

The text in Methods->Image quality->2nd paragraph now reads as follows:

Before MRIQC processing, a defaced mask was created for each subject, based on the NoMotion image, and applied to the motion simulated images. This removed pixels that were not present in the original masked dataset and thus facilitated a more correct comparison. Thereafter, all 148x7-8=1028 images were run through the MRIQC pipeline. Processing was performed using an MRIQC Docker container, version 23.1.0.

The corresponding section in the Discussion->2nd paragraph was also changed to:

The IQMs QI1 and QI2 showed inconclusive results across different (even real) nodding frequencies. It should be noted that the data was defaced making the air background-based analysis not possible [22].

And the sentence the reviewer refers to in the Discussion->6th paragraph now reads as follows:

Kemenczky et al. [33] used radiologist image quality rating scores, available in the published dataset, instead of IQMs to evaluate the motion robustness of deep learning-based brain segmentation tools compared to FreeSurfer.

R2.3 - In the Conclusion, the authors state that their simulation of realistic motion artifacts helps to mitigate biases between study groups and studies. There is no data in the paper to support this claim and it should be removed. Hopefully, the authors can apply their method and show such data in the future.

We thank the reviewer for their observation and have changed the relevant sentence as follows:

Future work could use this simulation approach to mitigate biases in the distribution of motion artifacts between study groups and to provide an unbiased foundation for the training of machine learning based models.

R2.4 - Although the proposed simulation more closely matches the real data in IQM distribution, they are still quite different, as seen in Fig. 5. This is especially true as the motion increases from 5 to 10 movements. The authors discuss limitations of their simulated data to match the real motion data. There are additional limitations, in addition to not being multi-channel k-space data or subjects having variable movement. For example, real motion can occur during different steps in the phase-encode along two axes. He MR-ART data uses an acceleration factor of GRAPPA=2. This type of real data can also have an interaction between the GRAPPA kernel used in the reconstruction and the motion corrupted k-space data. The authors could add a discussion about some of the other factors that produce artifacts in real motion corrupted MRI data, if they could be simulated in k-space based data, and if they could be simulated with magnitude MRI data.

A paragraph discussing the limitations of a magnitude data-based approach for motion simulations regarding parallel imaging (GRAPPA in particular), receive field sensitivity, translational (rather than rotational) motion was added to Discussion->5th paragraph:

Multi-channel phased arrays are typically employed for parallel imaging [36]. In the MR-ART dataset, a GRAPPA factor of 2 was applied, which means that every other k-space line is not measured but synthesized using weighting factors based on the measured data and indirect sensitivity measurements [37]. The interplay between the timing of discrete movements and the calculation of these weighting factors cannot be simulated using magnitude data alone. This is especially true when considering that receive sensitives are highly variable and depend on the position of the subject. Further, we have here focused only on simulating rotational motion where a rotation in image space results in an identical rotation in k-space. Translational motion will instead result in a linear phase ramp according to the Fourier shift theorem [38]. Although our suggested approach does not directly manipulate the pseudo k-space, it would be interesting to examine how a magnitude-based approach would compare to real-world data acquired when using a translational instead of rotational motion approach.

---

## [Decision Letter · Decision Letter 1]

5 Mar 2024

PONE-D-23-32888R1Simulating rigid head motion artifacts on brain magnitude MRI data – Outcome on image quality and segmentation of the cerebral cortexPLOS ONE

Dear Dr. Olsson, Thank you for submitting your manuscript to PLOS ONE. After careful consideration, we feel that it has merit but does not fully meet PLOS ONE’s publication criteria as it currently stands. Therefore, we invite you to submit a revised version of the manuscript that addresses the points raised during the review process. Please submit your revised manuscript by Apr 19 2024 11:59PM. If you will need more time than this to complete your revisions, please reply to this message or contact the journal office at plosone@plos.org. Please include the following items when submitting your revised manuscript:A rebuttal letter that responds to each point raised by the academic editor and reviewer(s). You should upload this letter as a separate file labeled 'Response to Reviewers'.A marked-up copy of your manuscript that highlights changes made to the original version. You should upload this as a separate file labeled 'Revised Manuscript with Track Changes'.An unmarked version of your revised paper without tracked changes. You should upload this as a separate file labeled 'Manuscript'.

We look forward to receiving your revised manuscript which I will be happy to accept upon your submission..

Kind regards,

Florian Ph.S Fischmeister

Academic Editor

PLOS ONE

Journal Requirements:

Reviewers' comments:

Reviewer's Responses to Questions

**Comments to the Author**

1. If the authors have adequately addressed your comments raised in a previous round of review and you feel that this manuscript is now acceptable for publication, you may indicate that here to bypass the “Comments to the Author” section, enter your conflict of interest statement in the “Confidential to Editor” section, and submit your "Accept" recommendation.

Reviewer #1: All comments have been addressed

Reviewer #3: All comments have been addressed

2. Is the manuscript technically sound, and do the data support the conclusions?

Reviewer #1: Yes

Reviewer #3: Yes

3. Has the statistical analysis been performed appropriately and rigorously? 

Reviewer #1: Yes

Reviewer #3: N/A

4. Have the authors made all data underlying the findings in their manuscript fully available?

Reviewer #1: Yes

Reviewer #3: Yes

5. Is the manuscript presented in an intelligible fashion and written in standard English?

Reviewer #1: Yes

Reviewer #3: Yes

6. Review Comments to the Author

Reviewer #1: The authors have addressed all the minor comments I had for their previous submission. I only have some minor suggestions:

- "a separate ethical approval was not obtained" instead of "a separate ethical approvement was not obtained"

- "For example, " or "e.g. " instead of "For an example subject"

- I would re-write this sentence:

"The pitch and duration of a single nod was empirically set to 15° and 2.5 s respectively. Based on the 5 s visual cue, it was assumed that each subject performed a nod within 0-5 s. Hence, a nod duration of 2.5 s was deemed reasonable."

to something like:

"The pitch magnitude was empirically set to 15°, while a nod duration of 2.5 s was deemed reasonable as it was assumed that each subject performed a nod within 0-5 s, based on a 5 s visual cue."

- I would re-write this sentence:

"BrainSuite does not directly output a cortical segmentation. A NIfTI volume that could be compared to the output of the other segmentation tools was therefore created using the mask of the boundary between white matter and cortical gray matter, a gray matter probability mask (pixels with >50% probability of belonging to GM was kept), and a mask of the cerebrum"

to something like:

"As BrainSuite does not directly output a cortical segmentation, we derived a NIfTI volume that could be compared to the output of the other segmentation tools by generating a mask of the boundary between white matter and cortical gray matter, as a gray matter probability mask (pixels with >50% probability of belonging to GM was kept), and a mask of the cerebrum".

- "The modified motion simulation described here has been applied to data from the ADNI (Alzheimer's Disease Neuroimaging Initiative)..." instead of "The modified motion simulation described here is currently being applied to data from the ADNI (Alzheimer's Disease Neuroimaging Initiative..."

- "using translational motion instead of rotation motion" instead of "using a translational instead of rotational motion approach".

Reviewer #3: This paper is very well structured, has clearly understandable language and is supported by the figures in the right places. The reviewers' comments were implemented in full.

Linguistically, I would recommend changing the first paragraph in the results section as a nuance: It would be more precise if instead of "ringing artifacts were represented" you would use "ringing artifacts were mimicked closer to the original" and instead of "instead showed a general blurring" you would rather use create or appeared as... (both starting from line 251).

7. PLOS authors have the option to publish the peer review history of their article (what does this mean?). If published, this will include your full peer review and any attached files.

Reviewer #1: No

Reviewer #3: No

---

## [Author Response · Author response to Decision Letter 1]

8 Mar 2024

Reviewer 1 (R1)

All the suggested phrase changes have been incorporated exactly as suggested. We appreciate the precise wording by the reviewer which made revision very convenient. One exception is the sentence below which was re-written to clearly state that three masks are used to create a single mask of the cortical gray matter:

As BrainSuite does not directly output a cortical segmentation, we derived a NIfTI volume that could be compared to the output of the other segmentation tools by combining three masks: A mask of the boundary between white matter and cortical gray matter, a mask of the total gray matter (cortical and deep gray matter) based on a GM probability map (pixels with >50% probability of belonging to GM was kept), and a mask of the cerebrum.

Reviewer 3 (R3)

R3.1 - Linguistically, I would recommend changing the first paragraph in the results section as a nuance: It would be more precise if instead of "ringing artifacts were represented" you would use "ringing artifacts were mimicked closer to the original" and instead of "instead showed a general blurring" you would rather use create or appeared as... (both starting from line 251).

The relevant sentences now reads as:

When performing the simulations (Fig 1), these ringing artifacts were mimicked closer to the real motion-corrupted images with the modified simulations (Mod5, Mod10). Images created with the original simulations (Ori5, Ori10) did not display these ringing artifacts as clearly but instead appeared as more blurred and with a worse overall image quality.

---

## [Editor Report · Decision Letter 2]

12 Mar 2024

Simulating rigid head motion artifacts on brain magnitude MRI data – Outcome on image quality and segmentation of the cerebral cortex

PONE-D-23-32888R2

Dear Dr. Olsson,

We’re pleased to inform you that your manuscript has been judged scientifically suitable for publication and will be formally accepted for publication once it meets all outstanding technical requirements.

Kind regards,

Florian Ph.S Fischmeister

Academic Editor

PLOS ONE
---

## [Editor Report · Acceptance letter]

1 Apr 2024

PONE-D-23-32888R2 

PLOS ONE

Dear Dr. Olsson, 

I'm pleased to inform you that your manuscript has been deemed suitable for publication in PLOS ONE. Congratulations! Your manuscript is now being handed over to our production team.

Kind regards, 

on behalf of

Mag. Dr. Florian Ph.S Fischmeister 

Academic Editor

PLOS ONE